# Gray's Anatomy for Segment Anything Model: Optimizing Grayscale Medical Images for Fast and Lightweight Segmentation

In Kyu Lee$^{1\star[0000-0001-5554-808X]}$, Jonghoe Ku$^{1*[0009-0003-1260-619X]}$, and YoungHwan Choi$^{2,3[0000-0003-4177-7724]}$

$^1$ Medipixel Inc, Seoul, Korea
$^2$ Department of Dermatology, Samsung Medical Center, Sungkyunkwan University School of Medicine, Seoul, Korea
$^3$ Department of Medical Device Management and Research, Samsung Advanced Institute for Health Sciences and Technology, Sungkyunkwan University, Seoul, Korea
Corresponding author: {soulkest}@skku.edu

**Abstract.** Advancements in medical image segmentation are critical for enhancing diagnostic accuracy in clinical settings, particularly when operating on edge devices like CPU-only laptops. In this context, we have developed a medical image segmentation model that is specifically designed for efficient deployment on such devices. Our approach leverages the EfficientViT-SAM architecture integrated with dynamic quantization to optimize both accuracy and computational efficiency. The model has been trained on a diverse dataset that includes over one million image-mask pairs from 10 different medical imaging modalities along with additional data for underrepresented anatomies. Performance evaluations show that our model achieves a dice score of 88.54% and a normalized surface dice of 98.28%, showing improvements of 4.37% and 2.85%, respectively, over the baseline model. The implementation of dynamic quantization not only preserves accuracy but also boosts inference speeds, making the model exceptionally viable for real-time clinical applications. This study affirms the potential of advanced segmentation technologies to operate effectively on non-specialized hardware, thereby expanding the accessibility of high-quality medical imaging analysis in environments constrained by resources. With its robust performance across various imaging scenarios and enhanced processing efficiency, the model promises substantial improvements in clinical workflows and patient outcomes. The code is available at https://github.com/Ninebell/GraysAnatomySAM.

**Keywords:** Medical imaging · Segmentation · Lightweight.

## 1 Introduction

Medical imaging plays a critical role in the diagnosis, treatment planning, and monitoring of various diseases. Segmentation, the process of delineating regions

---

$^\star$ Both authors contributed equally. Names are listed in alphabetical order.

of interest (ROIs) such as organs, lesions, and tissues in medical images, is fundamental to many clinical applications. Traditionally, manual segmentation has been the gold standard, but it is time-consuming, labor-intensive, and requires significant expertise. The advent of deep learning has brought substantial improvements, with models now capable of delivering accurate segmentation results across diverse tasks. However, these models are often task-specific, and their performance can degrade when applied to new tasks or different types of imaging data. This limitation has spurred interest in developing universal models for medical image segmentation that can generalize across various tasks.

The Segment Anything Model (SAM) [3] is a groundbreaking foundation model in the realm of image segmentation, demonstrating remarkable versatility and performance in natural image tasks. Despite its success, applying SAM directly to medical image segmentation presents challenges due to the inherent differences between natural and medical images. These differences necessitate adaptations to leverage SAM's strengths while addressing its limitations in the medical domain.

Recent studies have explored various adaptations of SAM to medical imaging, including MedSAM [6], which fine-tunes SAM on extensive medical image datasets to enhance its performance. Nonetheless, a significant barrier to the widespread adoption of these models in clinical settings is their computational intensity. SAM, in particular, requires substantial computational resources, making it impractical for time-sensitive and resource-constrained applications such as real-time diagnosis and mobile health applications.

To address this challenge, several lightweight versions of SAM have been proposed, such as FastSAM [11], MobileSAM [9], and EfficientSAM [8]. These models aim to reduce the computational burden while maintaining performance, often by employing techniques like model distillation and leveraging more efficient architectures. However, these adaptations still encounter trade-offs between performance and computational efficiency.

In this paper, we adopt EfficientViT-SAM [10] that combines the EfficientViT [2] architecture with SAM to create a fast and lightweight model for medical image segmentation. EfficientViT-SAM aims to retain the high performance of SAM while significantly reducing the computational requirements. In addition, we dynamically quantized our EfficientViT-SAM model for faster inference. Through comprehensive experiments on various medical imaging tasks, we demonstrate that our quantized EfficientViT-SAM achieves remarkable performance while being significantly faster and more efficient than existing models. Furthermore, we incorporate insights from recent research [4] on enhancing grayscale medical images to improve segmentation outcomes. By using the method we term "Gray's Anatomy", which processes grayscale medical images to optimize contrast and smoothness, we aim to boost the efficiency and accuracy of our model.

This work represents a step forward in making advanced medical image segmentation tools more accessible and practical for clinical use, highlighting the potential of combining foundation models with lightweight architectures and enhanced image preprocessing techniques.

## 2    Method

We introduce a comprehensive methodology for developing a lightweight medical image segmentation model. We start from preprocessing, including a three-channel enhancement technique that enriches input data essential for robust segmentation. Consequently, we elaborate on our adoption of the EfficientViT-SAM architecture, tailored for both accuracy and computational efficiency on edge devices. Finally, the post-processing subsection introduces dynamic quantization for fast and resource-efficient inference.

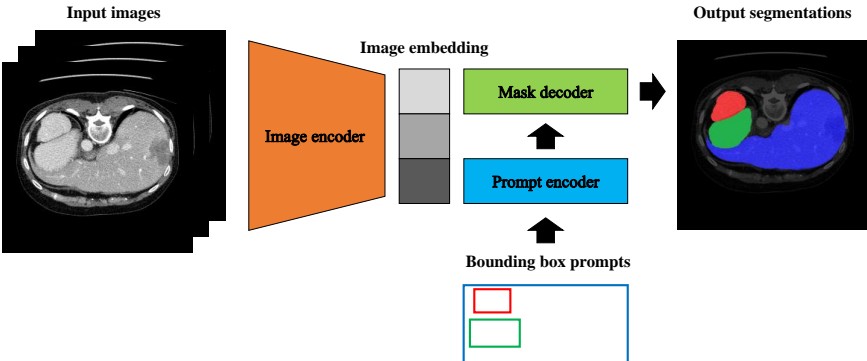

**Fig. 1.** Overview of the Image Segmentation Pipeline. The input is a three-channel constructed image, which is processed by the EfficientViT-SAM model to extract features across the entire image. Simultaneously, bounding box prompts specify regions of interest, guiding the model to focus its resources on specific areas. The final output is a segmentation mask applied within the regions defined by the prompts.

### 2.1    Preprocessing

**Data management**  Given the challenge of processing over a million image-mask pairs, data management and preprocessing are critical for efficient training and validation. To standardize the input data and optimize the use of computational resources, we preprocessed and saved data in a new format.

To ensure consistency across the dataset, which includes images of varying dimensions from different medical imaging modalities, we resized all images to have the same dimensions. Each image was first padded to make its height and width equal, preserving the aspect ratio and ensuring that no anatomical information was distorted or lost. Subsequently, we resized the images to a size

of 256x256 pixels. This resizing not only helps in maintaining computational efficiency but also ensures uniformity in the input data for our segmentation models.

To manage the large volume of segmentation masks accompanying the images, we employed run-length encoding (RLE) for the masks. RLE is a simple form of data compression where sequences of data (in this case, pixels of the same value) are stored as a single data value and count. This approach significantly reduced the size of our mask files, making them easier to store and faster to transmit.

The preprocessed images were saved in PNG format, which offers lossless compression, ensuring that no image data is lost after compression. The masks, encoded in RLE, were saved in Pickle (PKL) format, a Python-specific binary format, which facilitates easy loading and saving of large amounts of structured data. It was driven by the need for efficiency in both storage and speed during the training phase of our models.

**Channel Construction and Enhancement** The preprocessing stage is crucial for ensuring that the medical images are in an optimal format for segmentation while preserving essential diagnostic features. Considering the diverse characteristics of medical imaging modalities such as CT, X-ray, and MRI, which are primarily in a single-channel grayscale format, we adopted a multi-channel preprocessing approach to enhance segmentation accuracy and robustness.

Instead of replicating the grayscale image across three channels, we tailored each channel to capture different aspects of the image data, enhancing both the model's input variability and its capacity to identify relevant features. The first channel retains the original raw image data, serving as a baseline representation of the anatomical structure without any modifications. To reduce noise while preserving edge integrity, which is critical for delineating regions of interest, we applied anisotropic diffusion in the second channel. Anisotropic diffusion is particularly effective in environments with high levels of noise, as it smooths the image while maintaining sharp edges, crucial for accurate segmentation. The third channel builds upon the smoothed image from Channel 2. Here, we apply histogram equalization to maximize the contrast. This step is particularly beneficial for enhancing the visibility of subtle features within the image, which are often crucial for accurate and segmentation.

Figure 2 illustrates the transformation of a single grayscale image through these preprocessing steps, demonstrating the distinct contribution of each channel to the overall enhancement of the image. This preprocessing strategy not only standardizes the input data but also enriches the information the model receives, equipping it to more effectively differentiate between relevant features for segmentation tasks across various medical imaging modalities.

### 2.2   Proposed Method

**Architecture** To meet the challenge of developing a general and lightweight medical image segmentation model capable of running efficiently on laptops

without GPU support, we adopted the EfficientViT-SAM model, specifically its smallest variant, EfficientViT-SAM-L0. This architecture combines the strengths of scalable architecture modeling with an efficient version of the Vision Transformer, tailored for speed and low-resource consumption.

EfficientViT-SAM incorporates a hybrid approach that leverages the power of Vision Transformers while alleviating their traditional computational inefficiencies. The architecture begins with a series of MBConvolution (Mobile inverted Bottleneck Convolution) layers, which are specifically designed for mobile and edge devices due to their reduced parameter count and efficient computation. These layers preprocess the input image, effectively reducing its dimensionality while retaining crucial spatial hierarchies necessary for feature extraction. The processed features are then fed into the EfficientViT module. We initialized our image encoder with weights that are knowledge distilled from the SAM-ViT model. In this way, our EfficientViT-SAM image encoder retains the the robust feature recognition capability of more resource-intensive models while operating within the constraints of CPU-only environments.

Building on the EfficientViT-SAM's efficient feature extraction, the model incorporates MedSAM's bounding box prompt encoder and mask decoder. The bounding box prompt encoder enables the model to understand and process specific regions of interest within the image, focusing the segmentation task on areas highlighted by clinical relevance. The mask decoder then utilizes the features and spatial cues provided by the EfficientViT-SAM to generate precise segmentation masks, adapting dynamically to the varied shapes and sizes of medical anomalies.

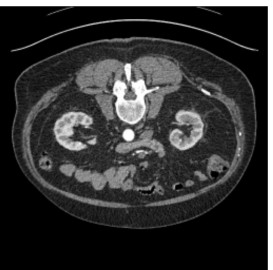 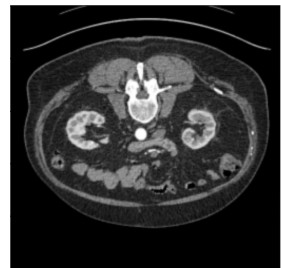 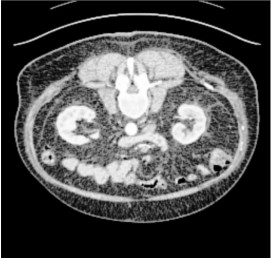

**Fig. 2.** Preprocessing Steps for Medical Image Segmentation. The left image displays the original raw grayscale image, showcasing the standard input format. The middle image illustrates the result of applying anisotropic diffusion, aimed at reducing noise while preserving critical edge details. The right image presents the histogram equalized image, where contrast has been enhanced to highlight subtle features and improve visibility, facilitating more accurate segmentation.

### 2.3   Post-processing

In the final stage of our model's deployment, we focus on optimizing the inference time and computational efficiency through the application of quantization techniques. Quantization reduces the precision of the numerical values in a model, which decreases the model's memory footprint and speeds up the processing time—essential traits for models intended for use on CPU-only laptops in clinical environments.

Quantization can be done in two ways: static quantization and dynamic quantization. Static quantization involves the conversion of both the weights and activations of the model to a lower precision format, such as int8, prior to deployment. This process requires the determination of quantization parameters, which are fixed during the initial calibration phase using a subset of the training data. While this method enhances the speed of operations by allowing the use of integer mathematics during inference, it suffers from a lack of flexibility. The predetermined scaling factors may not always accurately represent the range of values seen during actual model use, potentially leading to inaccuracies when processing data that differ significantly from the calibration dataset.

In contrast to static quantization, dynamic quantization offers a more adaptable solution for handling the variability inherent in medical imaging data. This approach involves quantizing the model's weights before deployment, while the activations are quantized dynamically at runtime. As a result, the quantization parameters for the activations are recalculated based on the actual data presented during inference. This dynamic adjustment allows the model to adapt to the specific characteristics of each image it processes, providing flexibility and accuracy crucial for medical applications where image diversity is high.

Opting for dynamic quantization using torch.qint8 enabled our model to maintain high segmentation accuracy while achieving substantial reductions in computational demands. This approach simplifies the deployment process by eliminating the need for extensive pre-calibration, thereby ensuring that the model can operate efficiently on a wide range of hardware, including the less powerful CPUs typical of many clinical settings. The dynamic nature of this quantization method enhances the model's usability and effectiveness, particularly in real-time clinical applications, making it a superior choice for ensuring robust performance across varied medical imaging scenarios.

## 3   Experiments

### 3.1   Dataset and evaluation measures

To comprehensively assess the performance of our segmentation model, we utilized a large-scale challenge dataset, which encompasses a diverse array of medical imaging modalities and cancer types. This dataset includes over one million image-mask pairs and covers 10 distinct imaging modalities, such as CT, MRI, and X-ray, providing a robust foundation for training our model. The dataset is also diverse in terms of anatomical coverage, featuring images of various body

parts including the lungs, skin, and eyes, which are critical for a wide range of clinical applications.

Recognizing the need to enhance our model's capability in handling less common anatomies and modalities, we supplemented the challenge dataset with external datasets. These additional datasets focus on anatomies and modalities not extensively covered in the challenge dataset, such as hip X-rays and ultrasound images of the prostate. The inclusion of these datasets ensures a more comprehensive training process, enabling our model to perform well across a broader spectrum of medical scenarios. Table 1 summarizes the external datasets incorporated into our training. This comprehensive approach to data collection allows our model to learn from a wide variety of image characteristics and clinical conditions, enhancing its generalizability in real-world applications.

To assess the performance of the segmentation models, this challege measured Dice Similarity Coefficient (DSC), Normalized Surface Dice (NSD) [1], and inference time as our primary evaluation metrics. DSC measures the volumetric overlap between the predicted segmentation and the ground truth, providing a quantitative indicator of the segmentation accuracy. It is used to evaluate the agreement between the two segmentations, where a value of 1 indicates perfect overlap and 0 indicates no overlap. In addition, NSD focuses on the accuracy of the segmentation boundaries rather than their volumetric correspondence. By measuring the similarity of the surfaces, NSD measures how well the segmentation contours align with the anatomical boundaries, which is crucial for applications requiring precise delineation of complex anatomical structures. Lastly, inference time is considered for the ranking computation. It ensures that they not only are accurate but also fit well within the operational constraints of medical environments.

**Table 1.** External training dataset.

| Dataset Name | Modality | Segmentation Targets | Annotated Images |
|---|---|---|---|
| Nuclei Segmentation | Microscopy | Nucleus | 5426 |
| HipXRay | X-ray | Bones | 140 |
| BTCV | CT | abdominal organs | 30 |
| Micro-Ultrasound | Micro-Ultrasound | Prostate | 75 |
| ToothSeg | X-ray | Teeth | 598 |

### 3.2   Implementation details

**Environment settings** The development environments and requirements are presented in Table 2.

**Training protocols** Training a model for medical image segmentation with over one million image-mask pairs presents unique challenges and constraints.

**Table 2.** Development environments and requirements.

| | |
|---|---|
| System | Windows 11 |
| CPU | AMD Ryzen 7 3700X 8-core Processor |
| RAM | 16×2GB; 2.67MT/s |
| GPU (number and type) | NVIDIA RTX 4090 24G |
| CUDA version | 11.0 |
| Programming language | Python 3.12 |
| Deep learning framework | torch 2.0, torchvision 0.2.2 |
| Specific dependencies | |
| Code | |

Our approach to training was carefully designed to optimize resource use while maintaining relevance to clinical applications.

Given the standardized nature of medical imaging and the critical importance of anatomical positions, conventional data augmentation techniques like flipping or random cropping are less suitable. For instance, anatomical landmarks such as the heart are consistently located in specific positions (e.g., the left side of the chest), making such transformations potentially misleading for a segmentation model. Therefore, our augmentation focused solely on adjusting the bounding box positions and sizes. This approach preserves the integrity and relevance of the anatomical information in the images, ensuring that the model learns to recognize and segment based on realistic variations in patient anatomy.

To evaluate our model effectively, we allocated 1% of the images from each modality to a validation set and reserved the remaining 99% for training. This split was designed to provide a robust dataset for training while ensuring that the validation set was representative of the diversity and challenges present in the larger dataset.

Due to the extensive size of our dataset and limitations in computational resources, completing even one epoch of training required more than a day. To manage this efficiently, we adopted a sampling strategy during training where only 1000 samples from each 2D imaging modality and 1000 samples from each 3D imaging submodality were used per epoch. This approach not only facilitated faster iterations but also made monitoring and saving model checkpoints more manageable. Model checkpoints were evaluated based on the performance on the validation set, with a focus on minimizing the validation loss. The model that demonstrated the smallest validation loss was selected for our final submission.

## 4   Results and discussion

### 4.1   Quantitative results on validation set

Our proposed model demonstrated significant enhancements in segmentation performance on the validation set compared to the baseline model, particularly in terms of DSC and NSD. Overall, improvements of 4.37% in DSC and 2.85%

**Table 3.** Training protocols.

| | |
|---|---|
| Pre-trained Model | Efficient-ViTSAM [10] |
| Batch size | 32 |
| Patch size | $256{\times}256{\times}3$ |
| Total epochs | 300 |
| Optimizer | AdamW [5] |
| Initial learning rate (lr) | 5e-5 |
| Lr decay schedule | ReduceLROnPlateau |
| Training time | 113.2 hours |
| Loss function | BCE, MSE, Dice |
| Number of model parameters | 34.79M[4] |
| Number of flops | 602G [5] |
| $CO_2$eq | 7 Kg[6] |

**Table 4.** Quantitative evaluation results.

| Target | Baseline | | Quantized Baseline | | Proposed | | Quantized Proposed | |
|---|---|---|---|---|---|---|---|---|
| | DSC(%) | NSD(%) | DSC(%) | NSD(%) | DSC(%) | NSD (%) | DSC(%) | NSD (%) |
| CT | 72.47 | 88.49 | 72.71 | 89.15 | 84.46 | 97.74 | 84.80 | 98.05 |
| MR | 76.40 | 93.02 | 77.12 | 93.22 | 80.92 | 95.86 | 82.69 | 97.22 |
| PET | 70.56 | 95.71 | 70.91 | 95.93 | 79.22 | 98.61 | 79.15 | 98.56 |
| US | 94.80 | 98.41 | 95.10 | 98.93 | 94.91 | 98.38 | 95.01 | 98.42 |
| X-Ray | 96.04 | 99.30 | 95.82 | 99.12 | 95.21 | 98.67 | 95.17 | 98.66 |
| Dermoscopy | 94.23 | 98.09 | 94.41 | 98.09 | 94.60 | 98.12 | 94.62 | 98.14 |
| Endoscopy | 91.47 | 98.46 | 91.45 | 98.34 | 91.38 | 98.60 | 91.40 | 98.57 |
| Fundus | 92.68 | 98.90 | 92.93 | 98.30 | 89.93 | 98.6 | 90.15 | 98.73 |
| Microscopy | 65.80 | 86.43 | 66.10 | 86.83 | 83.24 | 97.90 | 83.86 | 98.23 |
| Average | 83.83 | 95.20 | 84.06 | 95.32 | 88.20 | 98.05 | 88.54 | 98.28 |

in NSD were observed. While the baseline model showed stronger results in specific modalities such as X-ray, endoscopy, and fundus imaging, our proposed model excelled across a broader range of modalities, indicating its versatility and robustness. A significant improvement was observed in microscopy images, where the DSC dramatically increased from 65.80% to 83.24%. This substantial enhancement underscores the robustness of our model in handling various datasets.

Furthermore, the performance of both quantized versions of the baseline and the proposed models was evaluated to compare dynamic quantization. Remarkably, the quantized models did not exhibit a performance drop compared to their non-quantized counterparts, maintaining similar DSC and NSD scores. This result highlights the effectiveness of dynamic quantization as a post-processing step, confirming its potential to preserve the model's accuracy while significantly reducing the computational load during inference.

**Table 5.** Quantitative evaluation of segmentation efficiency in terms of running time (s).

| Case ID | Size | Num. Objects | Baseline | Proposed |
|---|---|---|---|---|
| 3DBox_CT_0566 | (287, 512, 512) | 6 | 210.98 | **51.85** |
| 3DBox_CT_0888 | (237, 512, 512) | 6 | 53.35 | **11.22** |
| 3DBox_CT_0860 | (246, 512, 512) | 1 | 7.50 | **2.51** |
| 3DBox_MR_0621 | (115, 400, 400) | 6 | 83.16 | **16.92** |
| 3DBox_MR_0121 | (64, 290, 320) | 6 | 51.20 | **9.84** |
| 3DBox_MR_0179 | (84, 512, 512) | 1 | 6.92 | **1.6** |
| 3DBox_PET_0001 | (264, 200, 200) | 1 | 3.50 | **0.78** |
| 2DBox_US_0525 | (256, 256, 3) | 1 | 0.40 | **0.11** |
| 2DBox_X-Ray_0053 | (320, 640, 3) | 34 | 0.90 | **0.72** |
| 2DBox_Dermoscopy_0003 | (3024, 4032, 3) | 1 | **1.41** | 2.09 |
| 2DBox_Endoscopy_0086 | (480, 560, 3) | 1 | 0.43 | **0.09** |
| 2DBox_Fundus_0003 | (2048, 2048, 3) | 1 | 0.72 | **0.38** |
| 2DBox_Microscope_0008 | (1536, 2040, 3) | 19 | 1.05 | **0.66** |
| 2DBox_Microscope_0016 | (1920, 2560, 3) | 241 | 6.27 | **6.19** |

### 4.2   Qualitative results on validation set

The qualitative evaluation of our model on the validation set further demonstrated its capability to accurately identify and segment regions of interest within bounding box prompts, particularly when a single region of interest is included within the prompt. For instance, as illustrated in the first and second rows of Figure 3, our model achieved precise segmentation of mammography and head CT images, respectively. In these cases, the bounding box prompts effectively covered the entire region of interest, allowing the model to correctly delineate the boundaries without interference from adjacent structures.

However, the model's performance was less consistent when faced with bounding box prompts containing multiple regions of interest. These scenarios often led to incorrect segmentations, as the model sometimes prioritized one region over another or misinterpreted the intended area of focus. Examples of such failures are displayed in the third and fourth rows of Figure 3. In one case involving an X-ray image of the lung, the model was prompted to segment the pneumothorax region but erroneously focused on the entire right lung. Another example includes a prompt covering both lungs and bones; here, the model incorrectly segmented the bone structure when the ground truth required segmentation of the entire lungs.

### 4.3   Segmentation efficiency results on validation set

An integral aspect of our evaluation focused on the efficiency of the segmentation process, particularly how dynamic quantization affects performance and inference speed. According to the results presented in Table 4, our dynamically quantized model maintained the accuracy of its non-quantized counterpart, with no degradation in DSC or NSD. This result underscores the effectiveness of dynamic quantization in preserving the integrity of the model's predictive capabilities while optimizing computational efficiency.

More importantly, the impact of quantization on inference speed was substantial. The dynamically quantized version of our proposed model demonstrated a remarkable increase in processing speed compared to the baseline model. Specifically, our proposed quantized model achieved up to five times faster inference times in certain scenarios and on average three times faster across all tested conditions.

These efficiency gains are further detailed in Table 5, which provides an overall comparison of inference times. This comprehensive overview underscores the significant speed advantages offered by our approach. Such improvements are particularly valuable in resource-constrained environments, aligning with our goal to develop a model that is both effective and efficient on edge devices.

### 4.4   Results on final testing set

We submitted our proposed model for evaluation using the final testing set. Overall, DSC and NSD metrics decreased by 9.6% and 16.6% respectively. Particularly in three-dimensional datasets, including CT, MR, and PET scans, our model performed poorly. The results from the final test set are presented in Table 4.

### 4.5   Limitation and future work

Our model faces challenges with bounding box prompts leading to segmentation ambiguities, such as whether to segment organs or bones in CT images. To improve clarity, future works could integrate more detailed prompting techniques,

**Image**        **Prediction**        **Ground truth**

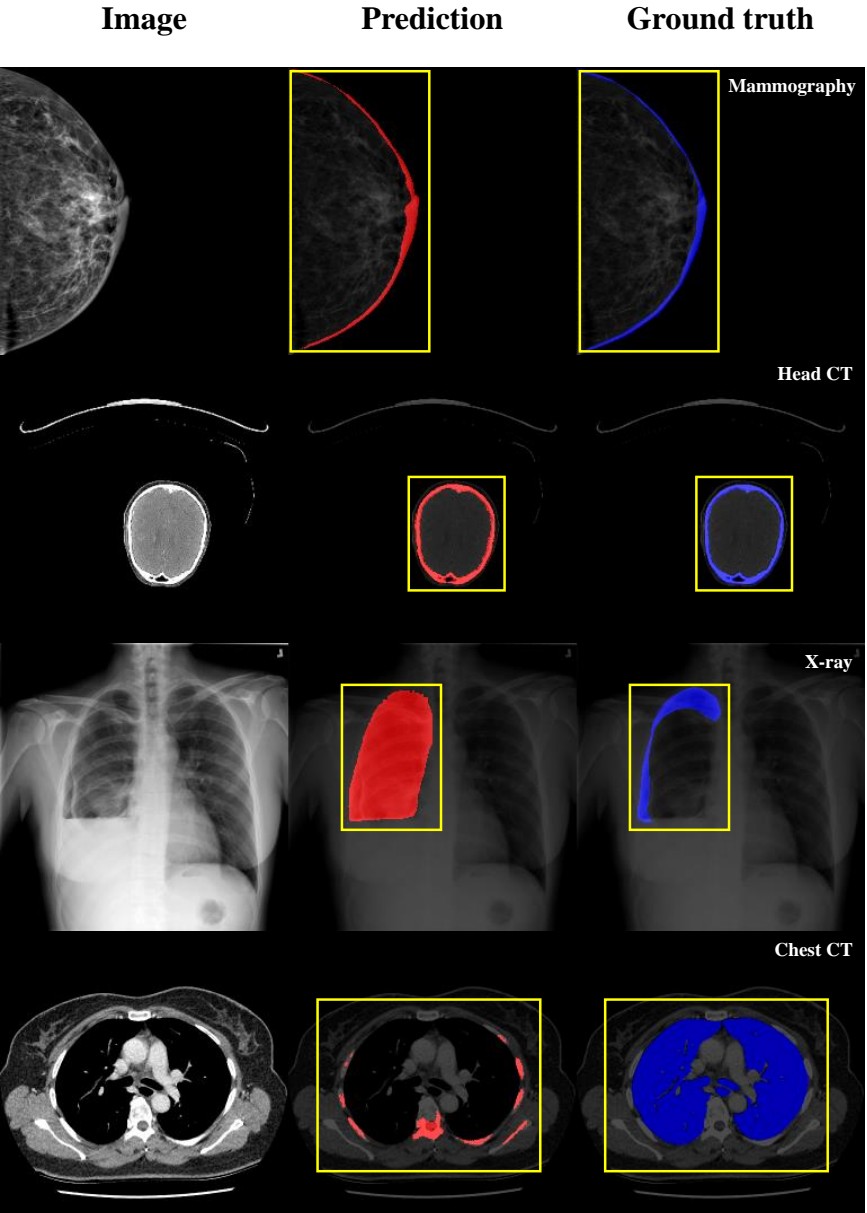

**Fig. 3.** Qualitative segmentation results. It illustrates examples of segmentation predictions made by our model. The top two rows display cases of successful segmentations. The third and fourth rows illustrate unsuccessful segmentations, where the predicted areas significantly diverge from the ground truth masks. Yellow bounding boxes represent prompts given to the model.

**Table 6.** Quantitative evaluation results on the test set.

| Target | Quantized Proposed DSC(%) | NSD (%) |
|---|---|---|
| CT | 62.63 | 65.49 |
| MR | 67.17 | 68.44 |
| PET | 77.73 | 68.50 |
| US | 86.02 | 90.42 |
| X-Ray | 74.16 | 85.62 |
| Endoscopy | 92.80 | 95.66 |
| Fundus | 90.61 | 92.64 |
| Microscopy | 82.14 | 84.43 |
| OCT | 77.30 | 84.39 |
| Average | 78.95 | 81.73 |

such as scribble-based input, where users provide direct annotations within the image, guiding the model more precise segmentation. Additionally, while our preprocessing methods successfully enhanced grayscale images, they proved less effective for color images. Future research can be focused on developing preprocessing techniques that improve feature recognition in color images as well, ensuring consistent performance across different imaging types. These refinements will boost the model's precision and expand its clinical utility.

## 5    Conclusion

In this study, we introduced a novel approach for medical image segmentation that leverages an efficient transformer-based architecture, EfficientViT-SAM, combined with dynamic quantization to achieve robust performance on edge devices, including laptops without dedicated GPU resources. Our methods addressed key challenges in medical imaging by providing a lightweight yet powerful solution capable of handling a diverse imaging modalities and anatomical structures.

Our results demonstrate that our proposed model significantly improves upon the baseline in terms of DSC and NSD, particularly showing notable performance enhancements in microscopy imaging where the segmentation accuracy increased dramatically. Importantly, the implementation of dynamic quantization ensured that these improvements did not come at the cost of computational efficiency. On the contrary, our quantized model achieved up to five times faster inference speeds, making it highly suitable for real-time clinical applications where rapid image processing is crucial.

Furthermore, our model's efficiency highlight its potential for widespread adoption in clinical settings, especially in scenarios where high computational resources are not available. This capability opens up new possibilities for deploying advanced medical imaging technologies in resource-limited environments, potentially enhancing patient care by providing quicker and more accurate diagnostic tools.

**Acknowledgements** We thank all the data owners for making the medical images publicly available and CodaLab [7] for hosting the challenge platform.

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

**Table 7.** Checklist Table. Please fill out this checklist table in the answer column.

| Requirements | Answer |
| --- | --- |
| A meaningful title | Yes |
| The number of authors ($\leq 6$) | 3 |
| Author affiliations and ORCID | Yes |
| Corresponding author email is presented | Yes |
| Validation scores are presented in the abstract | Yes |
| Introduction includes at least three parts: background, related work, and motivation | Yes |
| A pipeline/network figure is provided | Figure 1 |
| Pre-processing | Page 3 |
| Strategies to data augmentation | Page 4 |
| Strategies to improve model inference | Page 5-6 |
| Post-processing | Page 6 |
| Environment setting table is provided | Table 2 |
| Training protocol table is provided | Table 3 |
| Ablation study | Page 9-10 |
| Efficiency evaluation results are provided | Table 5 |
| Visualized segmentation example is provided | Figure 3 |
| Limitation and future work are presented | Yes |
| Reference format is consistent. | Yes |
| Main text $>= 8$ pages (not include references and appendix) | Yes |