# OpenReview forum: "Gray’s Anatomy for Segment Anything Model: Optimizing Grayscale Medical Images for Fast and Lightweight Segmentation"
_thecvf.com/CVPR/2024/Workshop/MedSAMonLaptop — CVPR24 MedSAMonLaptop_

### Official Review · Reviewer_qtLY · 2024-06-12
**Efficienct pre-processing of the MedSAM training data combined with dynamic quantization of the EfficientViT-SAM model.**

**Rating:** 8
**Confidence:** 4

**Review:**

The authors utilize the EfficientViT-SAM backbone combined with dynamic quantization to improve the efficiency of the LiteMedSAM baseline. They replace the 3 duplicate channels used for grayscale images with anisotropic diffusion and histogram equalization to enhance well-pronounced edge features directly in the input image. The authors also pre-process MedSAM's provided training dataset to a fixed resolution of 256 x 256 and encode the binary masks with Run-Length-Encoding to save space on the disk.

The paper is well-written and the methodology is sound and complete. The authors explain everything in enough detail so that is possible to reproduce their results and model training. I only found one small comment: Table 1 should include a reference to each dataset. Other than that, the paper seems to be complete and reproducible.

Hence, I would opt for an acceptance of the manuscript.

---

### Official Review · Reviewer_cxKc · 2024-06-13
**CPU-optimized medical image segmentation model using EfficientViT-SAM, channel enhancement and dynamic quantization**

**Rating:** 7
**Confidence:** 5

**Review:**

The paper discusses the development of a medical image segmentation model optimized for deployment on CPU-only devices, such as laptops in light of the "CVPR 2024: Segment Anything In Medical Images On Laptop" challenge . This model leverages the EfficientViT-SAM architecture integrated with dynamic quantization to balance accuracy and computational efficiency. It furthermore uses a three-channel enhancement technique for grayscale images - retaining the raw image, applying anisotropic diffusion, and performing histogram equalization. The model was trained on over a million image-mask pairs from ten different medical imaging modalities provided by the organisers and demonstrated improvements in performance metrics, such as a 4.37% increase in dice score and a 2.85% increase in normalized surface dice compared to the baseline model, with faster inference speed for nearly all modalities.


Strengths:

Efficiency and Performance: The model achieves a segmentation accuracy with a reported dice score of 88.54% and a NSD of 98.28% outperfroming the baseline. Remarkably, the use of dynamic quantization not only boosts inference speed by up to five times but also slightly improves or maintained performance.

Innovative Preprocessing: The three-channel enhancement technique for grayscale images improves the model's ability to handle diverse imaging modalities and enhances segmentation accuracy.

Weaknesses:

Ablation Study: An ablation study without the three-channel sampling and with just the dynamic quantization would clarify their individual contributions.

Preprocessing Explanation: The explanation of the three-channel preprocessing is insufficient. While citing a paper in the introduction section, more detail in the methods section would improve reproducibility and understanding, especially since this is a key contribution to the challenge.

Segmentation Ambiguities: The model sometimes struggles with bounding box prompts that contain multiple regions of interest, leading to segmentation inaccuracies. This limitation is acknowledged but not thoroughly addressed in the study.

Metrics: Based on the public validation leaderboard, the reported NSD value of 98.28% appears unusually high. The top-performing method currently achieves an NSD of 89.16%, which is significantly lower than the reported figure.


Reproducibility:

Although the authors mention that the code will be available at a specified link, it has not been made public yet. Additionally, the method description lacks some details that could be improved for better reproducibility.

---

### Official Review · Reviewer_vK1L · 2024-06-13
**Review of "Gray’s Anatomy for Segment Anything Model: Optimizing Grayscale Medical Images for Fast and Lightweight Segmentation"**

**Rating:** 7
**Confidence:** 4

**Review:**

The paper introduces and briefly discusses three main additions to the (Lite)MedSAM baseline:
* **Altered Model Architecture**: The authors replace the image encoder of the baseline model with EfficientViT-SAM-L0.
* **Increased Speed through Quantization**: They explore static and dynamic quantization, demonstrating benefits such as reduced inference time while maintaining accuracy compared to the non-quantized model.
* **Enhancing Input for Grayscale Images**: For single channel modalities, they show that applying transformations to the original input image and using those altered image as further input improves segmentation results compared to just duplicating the grayscale image over all 3 input channels.

Overall, the paper is clearly structured and understandable and checks all boxes in table 6. The authors plan to publish the model's code and have disclosed their training data, making it feasible to reproduce their experiments.

Positives:
* They show that a more lightweight image encoder can be used in the SAM framework.
* Simple yet effective enhancement for improving image segmentation quality for grey-scale images.

Negatives:
* The baseline scores from Table 4 do not match with the reported numbers here https://www.codabench.org/forums/1766/247/
  However it is not clear whether the baseline was tested on the validation set or on their hold-out part of the training set. Clarification would increase trust in the paper.

---

### Official Review · Reviewer_W5Zq · 2024-06-13
**Review of "Gray’s Anatomy for Segment Anything Model: Optimizing Grayscale Medical Images for Fast and Lightweight Segmentation"**

**Rating:** 7
**Confidence:** 4

**Review:**

Summary:

The paper describes the development of a segmentation network, designed in the light of the “CVPR 2024: Segment Anything In Medical Images On Laptop” Challenge. The submission lists three main contributions - enhancing grayscale images to optimize contrast and smoothness, leveraging the EfficientViT-SAM architecture and employing dynamic quantization. The authors show that this combination leads to improved performance and reduced inference speed compared to the baseline model on their heldout validation set.

Strengths:
- Preprocessing: The authors apply a novel preprocessing scheme to utilize the channel information for grayscale images.
- Performance: The proposed method improves significantly upon the baseline.
- Efficiency: The combination of the EfficientViT-SAM encoder and the use of dynamic quantization lead to a reduction of inference time of up to 80%.

Weaknesses:
- Missing details: The paper lacks many important details, which reduce reproducibility of the submission.
  - It is not completely clear how the model was trained, especially regarding knowledge distillation. Which checkpoint was used for knowledge distillation?
  - The grayscale enhancement is described, but it is not clear how a combination of this choice of channel input and the channel inputs for RGB images should be done. It seems like this actually leads to a slight decrease in performance of modalities with RGB input.
- Validation Set: The authors report results on a custom 1% validation split instead of the full training set instead of the official validation split provided by the organizers. This not only makes a comparison to other submissions basically impossible, but also the comparison to the baseline model LiteMedSAM, which was trained on a large portion of this data. The custom validation set seems to show significant differences to the official validation set, as can be seen by e.g. the performances on CT and MR.
- Results: The presented results for NSD are questionable. A reported NSD of ~0.98 with a DSC of 0.88 seems to be too high. Especially compared to results on the official validation set, these values appear to be way too high and raise questions regarding a correct implementation of the evaluation metrics.
- Dynamic Quantization: It seems like dynamic quantization actually leads to an improvement of performance compared to the unquantized network. As this is very unexpected, the authors should mention this and if possible try to give an explanation.
- Ablation study: It is not clear how the different contributions contributed to the final performance. What is the contribution of the grayscale enhancement on top of training on more data?

---

### Decision · Program_Chairs · 2024-10-01

Accept